# Epidemiological Transitions in Influenza Dynamics in the United States: Insights from Recent Pandemic Challenges

**DOI:** 10.3390/microorganisms13030469

**Published:** 2025-02-20

**Authors:** Marta Giovanetti, Sobur Ali, Svetoslav Nanev Slavov, Taj Azarian, Eleonora Cella

**Affiliations:** 1Department of Sciences and Technologies for Sustainable Development and One Health, Università Campus Bio-Medico di Roma, 00128 Roma, Italy; giovanetti.marta@gmail.com; 2⁠Oswaldo Cruz Institute, Oswaldo Cruz Foundation, Belo Horizonte 30190-002, MG, Brazil; 3Climate Amplified Diseases and Epidemics (CLIMADE), Belo Horizonte 30190-002, MG, Brazil; 4Burnett School of Biomedical Sciences, College of Medicine, University of Central Florida, Orlando, FL 32827, USA; mdsobur.ali@ucf.edu (S.A.); taj.azarian@ucf.edu (T.A.); 5Butantan Institute, São Paulo 05585-000, SP, Brazil; svetoslav.slavov@fundacaobutantan.org.br

**Keywords:** influenza, flu, SARS-CoV-2, epidemiology, shift, US

## Abstract

The SARS-CoV-2 pandemic has reshaped the epidemiological landscape of respiratory diseases, with profound implications for seasonal influenza. Nonpharmaceutical interventions implemented globally during the pandemic significantly altered human behavior and reduced the prevalence of respiratory pathogens, including influenza. However, the post-pandemic resurgence of influenza activity to pre-pandemic levels highlights the persistent challenges posed by this virus. During the 2023–2024 influenza season in the United States, an estimated 40 million individuals contracted influenza, resulting in 470,000 hospitalizations and 28,000 deaths, with the elderly disproportionately affected. Pediatric mortality was also notable, with 724 deaths reported among children. This study examines trends in influenza incidence, vaccination rates, and mortality in the United States from the 2018–2019 through to the 2023–2024 influenza seasons. Additionally, it evaluates the interplay between influenza and SARS-CoV-2 during the pandemic, considering the impact of disrupted air travel, public health measures, and altered virus circulation dynamics. By integrating these insights, the study underscores the critical need for sustained vaccination campaigns and innovative public health strategies to mitigate the dual burden of respiratory diseases. Findings from this analysis highlight the urgency of strengthening prevention and surveillance systems to enhance pandemic preparedness and reduce the impact of respiratory pathogens in an evolving epidemiological landscape.

## 1. Introduction

The SARS-CoV-2 pandemic has profoundly impacted human health, both directly and indirectly. During the pandemic, the widespread implementation of nonpharmaceutical interventions (NPIs), such as social distancing, mask use, isolation, and quarantine, led to significant changes in human behavior [1]. These measures resulted in a remarkable decrease in the global prevalence of epidemic respiratory pathogens, including seasonal influenza [2].

In the 2023–2024 influenza season, flu severity was classified as moderate, with activity levels returning to pre-pandemic patterns [3,4]. According to the CDC, an estimated 40 million individuals in the United States contracted influenza, with 18 million seeking medical care, 470,000 hospitalized, and 28,000 fatalities. Elderly patients (≥65 years) were disproportionately affected, accounting for 68% of deaths and 50% of hospitalizations [5]. Pediatric mortality also remained significant, with 724 estimated deaths among children and neonates. These findings underscore the urgent need for comprehensive vaccination campaigns featuring vaccine compositions tailored to the circulating strains, alongside other preventive measures, such as widespread public health communication campaigns and the implementation of NPI, to mitigate the public health burden of seasonal influenza [6]. Unfortunately, the pandemic also indirectly impacted immunization rates for other vaccine preventable diseases, including influenza [7,8].

The pandemic disrupted the global circulation of seasonal influenza, partly due to changes in air travel connectivity between world regions [9]. Moreover, the initial decline in flu activity was strongly influenced by the global adoption of stringent public health measures, including mask mandates and physical distancing [10,11,12], which effectively curtailed the person-to-person transmission of respiratory viruses. In the US, flu activity during the 2020–2021 season was the lowest it had been since the CDC began its current reporting system in 1997 [13].

Since the emergence of SARS-CoV-2 in December 2019, it has infected over 700 million individuals globally and caused more than seven million deaths. The United States alone was one of the most affected countries, with CDC estimates bordering 40 million COVID-19 infections occurring up to 2024 [14]. Several factors contributed to the U.S.’s high case burden, including its large population, delayed response measures, healthcare strain, high rates of underlying conditions, and social disparities. These factors resulted in one of the highest case and death tolls globally, despite containment efforts.

This study aims to provide a comprehensive analysis of influenza incidence, vaccination rates, and mortality in the United States across three critical phases: before, during, and after the COVID-19 pandemic. By examining data from the 2018–2019 through to the 2023–2024 influenza seasons, we elucidate how the interplay between influenza and SARS-CoV-2 has reshaped public health strategies and underscore the urgent need for sustainable interventions to mitigate the dual burden of respiratory diseases in the post-pandemic era.

## 2. Materials and Methods

The influenza surveillance period follows a seasonal framework, spanning 52 or 53 weeks, beginning in week 36 of the current year and ending in week 35 of the subsequent year. This structure allows for the consistent tracking and analysis of influenza trends over time. For this study, data from the 2018–2019 through to the 2023–2024 influenza seasons were retrieved from the CDC’s FluView repository (https://www.cdc.gov/fluview/index.html, accessed on 15 December 2024), providing a comprehensive overview of influenza activity across six consecutive seasons. To investigate the interplay between COVID-19 and influenza dynamics during the pandemic, additional data on COVID-19 incidence from 2020 to 2024 were obtained from both the CDC repository and the global data platform “Our World in Data” (https://ourworldindata.org/) (specifically for the containment and health index). This dataset enabled a detailed exploration of the temporal and spatial fluctuations in influenza activity, alongside the emergence and spread of SARS-CoV-2. By integrating these datasets, the study aims to capture the impact of the pandemic on the circulation and incidence of seasonal influenza and to identify potential shifts in respiratory disease patterns influenced by public health interventions and changing societal behaviors. Interrupted time series (ITS) analysis was conducted to assess changes in weekly influenza case trends between the flu seasons in pre-pandemic, pandemic, and post-pandemic time periods. The Wilcoxon signed-rank test was used to evaluate the change in flu cases and flu vaccination coverage between flu seasons in the U.S. states. A *p*-value < 0.05 was considered statistically significant. Data analysis and visualization were carried out in R studio v2024.04.2, and R package v4.4.1 was used for map visualization.

## 3. Results

The analysis of weekly influenza case data spanning six influenza seasons (2018–2019 to 2023–2024) revealed notable temporal patterns influenced by the emergence and global spread of SARS-CoV-2 (Figure 1).

During the pre-pandemic period (2018–2019 to 2019–2020), influenza cases followed predictable seasonality, with activity typically rising in late autumn, peaking in mid-winter, and tapering off by early spring. This cyclical pattern was disrupted during the COVID-19 pandemic (2020–2022), with influenza activity experiencing an unprecedented decline. Influenza cases were markedly suppressed in the 2020–2021 flu season during the early pandemic years, coinciding with the implementation of widespread nonpharmaceutical interventions (NPIs), such as mask-wearing, physical distancing, and international travel restrictions. It is worth noting that there was a decrease in testing for influenza and other viruses, like dengue [15], during the COVID-19 pandemic, since there was a shift through its testing, which might have led to the under-reporting of cases [16,17]. The containment and health index, plotted alongside influenza cases, reflects the intensity of these measures, which peaked in early 2020. However, there was the first initial peak during the pandemic that coincided approximately with the relaxation of NPI practiced during the SARS-CoV-2 pandemic. In the last year of the pandemic (flu season 2022–2023) and post-pandemic period (flu season 2023–2024), influenza activity exhibited a robust rebound, with higher peaks compared to the pre-pandemic era (Appendix A). This resurgence underscores the waning impact of NPIs, as public health measures were relaxed and global travel resumed. The results highlight how behavioral and policy-driven interventions, though effective in the short term, are not sufficient to sustain the long-term suppression of respiratory pathogens like influenza.

The geographic distribution of influenza cases across U.S. states over six influenza seasons illustrates the differential impact of the pandemic and subsequent recovery periods across regions (Figure 2). Additionally, a correlation analysis indicated that states in the U.S. with higher flu incidence experienced more COVID-19 cases during (season 2020–2021) and after the pandemic period (season 2021–2022 and 2022–2023) (*p* < 0.001, Appendix A). This indicates that common risk factors, including population density, healthcare accessibility, and behavioral determinants, may affect the transmission dynamics of these respiratory viruses [18].

Pre-pandemic seasons (2018–2019 and 2019–2020) show widespread influenza activity, with case hotspots appearing consistently across highly populated federal units—such as California, Texas, and New York. However, during the pandemic (2020–2021), influenza transmission was significantly reduced nationwide, with no state showing significant activity (*p* < 0.001, Appendix A). This decline aligns with reduced person-to-person contact, restricted mobility, and the widespread adoption of masking policies. The 2021–2022 and 2022–2023 seasons marked the significant resurgence of influenza (*p* < 0.001, Appendix A), with case clusters re-emerging in several states. Interestingly, the geographic distribution during these seasons suggested a shift in hotspot locations, potentially influenced by varying levels of public health compliance, vaccine uptake, and demographic factors [19,20]. By the 2023–2024 season, influenza activity was widely distributed again, emphasizing the adaptive nature of the virus and the need for sustained regional surveillance and targeted interventions.

Influenza-related deaths mirrored the trends observed in case counts, with a pronounced reduction following the pandemic years (2021–2022) and a gradual return to pre-pandemic levels in subsequent seasons (Figure 3). During the 2018–2019 and 2019–2020 seasons, influenza mortality was widespread, with higher death tolls observed in densely populated states and regions with older demographics. The 2021–2022 season showed a dramatic decline in mortality, reflecting the significant suppression of influenza transmission during this period. In the 2022–2023 seasons, influenza-related mortality began to increase again, coinciding with the resurgence in cases. However, geographic patterns of mortality showed some variation, likely influenced by differences in healthcare access, vaccination coverage, and the prevalence of underlying health conditions. Mortality data for the 2020–2021 (due to SARS-CoV-2 mortality) and 2023–2024 season were unavailable at the time of this analysis, limiting the ability to fully assess recent trends. These findings underscore the critical role of effective surveillance, timely intervention, and vaccination campaigns in minimizing the burden of influenza-related deaths. The observed decrease in influenza mortality and case counts during the pandemic can be attributed to the implementation of effective surveillance systems, timely public health interventions, and robust vaccination programs. These measures enabled early detection and response to influenza outbreaks, thereby mitigating the spread of the virus and reducing the overall impact on public health.

Influenza vaccination coverage varied substantially across states and seasons, reflecting differences in public health strategies, population behaviors, and vaccine acceptance (Figure 4). Pre-pandemic seasons (2018–2019 and 2019–2020) exhibited moderate vaccination uptake, with mean coverages of 73.9% and 76.5%, respectively, across states. During the early pandemic period (2020–2021 flu season), vaccination rates significantly increased (mean: 84.6%, *p* < 0.001) compared to the pre-pandemic period (Appendix A), likely due to heightened public awareness of respiratory diseases and enhanced public health messaging.

However, in the post-pandemic seasons (2022–2024), vaccination coverage exhibited considerable variability across states, with some regions showing declines in uptake. Despite the increased risk of co-circulating respiratory pathogens (i.e., respiratory syncytial virus, SARS-CoV-2, *Streptococcus pneumoniae*, human metapneumovirus), vaccination rates remained below optimal levels in several states, particularly in the southern and midwestern regions. However, there was a statistically significant increase in vaccination rates pre and decrease in post pandemic seasons scattered across some U.S. states (*p* < 0.001, Appendix A), though not uniformly. These patterns highlight the challenges of maintaining public interest and trust in vaccination campaigns after the acute phase of a pandemic has passed. Importantly, sustained and equitable vaccination efforts are necessary to mitigate the long-term impact of influenza and other vaccine-preventable respiratory pathogens.

## 4. Discussion

The SARS-CoV-2 pandemic profoundly impacted human health on multiple levels. Beyond the direct effects of viral exposure, which included high mortality rates [14] and chronic conditions, such as long COVID-19 [21,22,23], the pandemic indirectly disrupted the global epidemiology of respiratory pathogens, including the seasonal circulation of influenza strains. For three years, the pandemic-induced perturbation significantly altered the global dispersion of influenza viruses; although, by 2023, the observed pattern reverted to its pre-pandemic state. This phenomenon underscores the impact of public health interventions and behaviors on respiratory virus dynamics [24,25]. During the pre-pandemic period, influenza activity exhibited predictable seasonal patterns, peaking during mid-winter in highly populous states, such as California, Texas, and New York. However, during the 2020–2021 season, the widespread implementation of NPIs, such as mask-wearing, physical distancing, and travel restrictions, led to an unprecedented suppression of influenza cases across the United States. Another critical factor that must also be considered is the redirection of diagnostic efforts toward detecting SARS-CoV-2 in clinical samples on a large scale, which has likely contributed to the underestimation of influenza prevalence and strain circulation.

Similar trends were observed globally, in the US, Australia, Italy, Chile, South Korea, and Chile [26,27,28,29,30,31,32,33,34,35]. This also led to the underestimation of the circulation of other respiratory viruses, including respiratory syncytial virus (RSV), among others, during the first couple of years of the pandemic [32,36,37,38]. However, an underestimation of other pathogens happened due to the extensive focus on SARS-CoV-2 diagnosis [15]. The effectiveness of these measures was reflected in the peak of the containment and health index in early 2020 [39]. However, the suppression of influenza activity was not largely sustained. With the easing of the restriction measures, the return of global mobility, and effective influenza testing, an increase in influenza activity during the 2021–2023 seasons was observed, eventually returning to pre-pandemic levels in 2023–2024. Moreover, there was a significant improvement in the availability of diagnostic platforms, such as BioFire, which enhanced the ability to quickly and accurately detect various pathogens. This advancement played a crucial role in managing the spread of infectious diseases by enabling timely diagnosis and appropriate treatment [40,41]. This resurgence underscores the adaptive nature of influenza viruses and the challenges of relying on restrictive containment measures as long-term suppression strategies post-pandemic.

Geographic analysis during the post-pandemic period revealed a shift in influenza hotspots, influenced by regional disparities in public health compliance, vaccine administration, and demographic factors [26,32,42,43]. Regions with lower vaccination coverage, particularly in the southern and midwestern states, experienced variability in the incidence of influenza activity during the flu seasons. Although public health campaigns during the pandemic temporarily increased vaccination rates due to heightened awareness [43,44,45], this momentum waned in the post-pandemic period, with some states exhibiting significant declines. Influenza-related mortality mirrored these trends, with a dramatic reduction in 2021–2022, followed by a gradual rise in the subsequent season [32]. Variations in healthcare access, vaccination coverage, and comorbid conditions likely contributed to the geographic disparities in mortality, consistent with prior research [46,47,48,49].

To mitigate the burden of influenza in future seasons, a comprehensive and sustained approach is essential. Increasing vaccination coverage remains the cornerstone of influenza control, with efforts needed to address vaccine hesitancy and logistical challenges, particularly in underserved regions [44,50]. Advances in genomic surveillance and predictive modeling offer opportunities to refine intervention strategies, enabling real-time monitoring and response to emerging threats [51,52,53,54,55]. Public health strategies should also incorporate lessons from the pandemic, combining targeted NPIs and advanced diagnostic strategies with timely vaccination campaigns during high-risk periods to reduce influenza transmission, while minimizing societal disruption [26,27,56].

## 5. Conclusions

The concurrent analysis of influenza and SARS-CoV-2 dynamics during the study period revealed significant interactions between these two respiratory pathogens. The sharp decline in influenza cases during the pandemic years suggests both direct and indirect effects. Directly, viral interference may have contributed to SARS-CoV-2 outcompeting influenza for susceptible hosts. Indirectly, NPIs, such as masking, physical distancing, and travel restrictions, significantly suppressed influenza transmission. However, as public health measures were relaxed, influenza re-emerged, demonstrating the virus’s inherent adaptability and resilience.

The resurgence underscores the adaptive nature of influenza viruses and the challenges of relying on restrictive containment measures as long-term suppression strategies post-pandemic. The observed patterns highlight the critical role of nonpharmaceutical interventions (NPIs), altered virus circulation dynamics, and vaccination efforts in shaping respiratory virus epidemiology. Moreover, it highlights the necessity of integrating surveillance systems for multiple respiratory pathogens. A comprehensive understanding of co-circulating viruses is crucial for developing effective public health strategies that minimize the burden of respiratory illnesses and enhance preparedness for future pandemics.

## Figures and Tables

**Figure 1 microorganisms-13-00469-f001:**
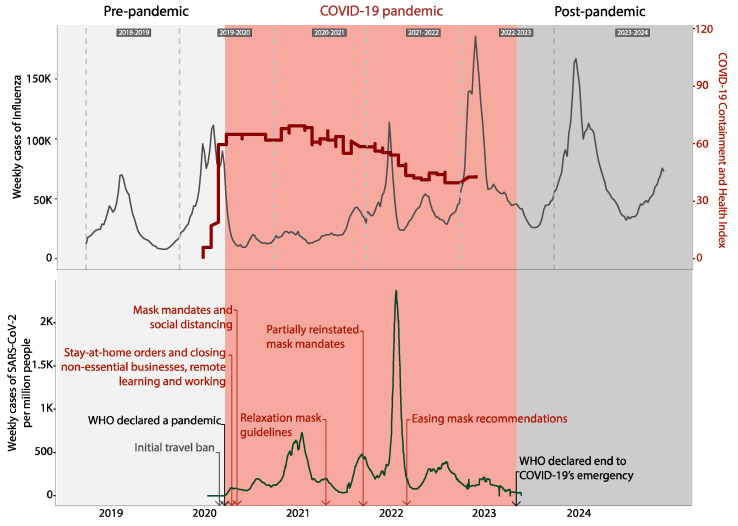
Seasonal influenza and SARS-CoV-2 dynamics before, during, and after the COVID-19 pandemic. Trends in weekly reported cases of influenza (top panel) and SARS-CoV-2 (bottom panel) in the United States from the 2018–2019 to 2023–2024 seasons. The graph is divided into three distinct periods: pre-pandemic (2018–2019), COVID-19 pandemic (2020–2022, shaded in red), and post-pandemic (2023–2024, shaded in gray). The red line in the top panel indicates the COVID-19 Containment and Health Index (obtained from https://ourworldindata.org/), reflecting the intensity of nonpharmaceutical interventions implemented during the pandemic. The gray vertical lines delimit the different influenza seasons. Major federal containment initiatives for SARS CoV-2 are showed in the bottom panel.

**Figure 2 microorganisms-13-00469-f002:**
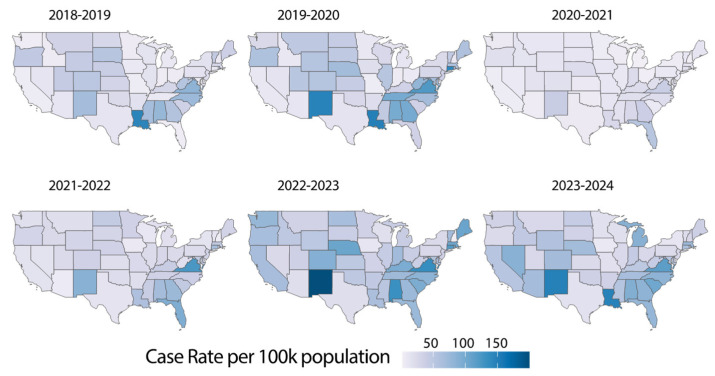
Geographic trends in influenza cases across six seasons (2018–2024). Heatmaps of influenza cases rate per 100,000 population across U.S. states for the 2018–2019 to 2023–2024 seasons. The 2020–2021 season shows a marked decline in activity during peak COVID-19 measures, while later seasons display a return to pre-pandemic levels, emphasizing regional variability and public health impact. States with low cases or missing data are colored in white.

**Figure 3 microorganisms-13-00469-f003:**
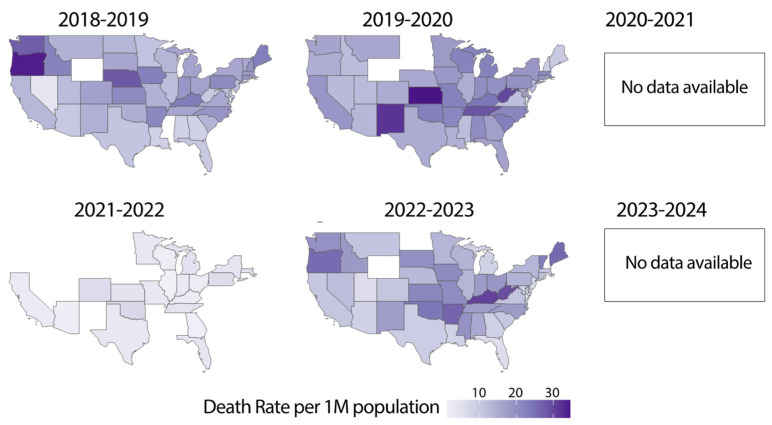
Geographic distribution of influenza-related deaths in the United States (2018–2024). Heatmaps showing influenza-related death rate per 1M population across U.S. states during the 2018–2019 to 2022–2023 influenza seasons. Data for the 2020–2021 and 2023–2024 seasons were unavailable. Darker shades represent higher mortality rates, with notable variability in the geographic burden of influenza-related deaths. The absence of data for certain seasons highlights the challenges in consistent surveillance and reporting during and after the COVID-19 pandemic. States with low deaths or missing data are colored in white.

**Figure 4 microorganisms-13-00469-f004:**
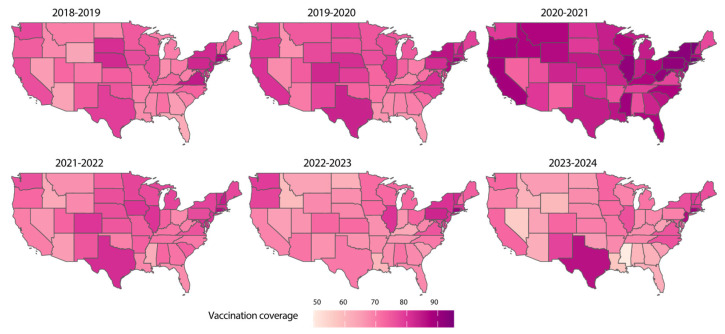
Seasonal influenza vaccination coverage across the United States (2018–2024). Heatmaps illustrating vaccination coverage for influenza across U.S. states during the 2018–2019 to 2023–2024 influenza seasons. Darker shades represent higher vaccination coverage, with variations observed between seasons and regions. The 2020–2021 season, marked by the COVID-19 pandemic, shows a moderate increase in vaccination coverage compared to prior seasons, likely influenced by heightened public health awareness. Subsequent seasons demonstrate variability in coverage, emphasizing the need for sustained vaccination efforts to achieve optimal influenza prevention. States with missing data are colored in white.

## Data Availability

The data used in this study are freely available in CDC data website and are available upon request from the corresponding author.

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
