# Peer review of "Epidemiological Transitions in Influenza Dynamics in the United States: Insights from Recent Pandemic Challenges"

_microorganisms, 2025, doi:10.3390/microorganisms13030469_

Round 1

Reviewer 1 Report

Comments and Suggestions for Authors

This is an important article. However, it lacks appropriate statistical analyses.

You wrote: “The influenza surveillance period follows a seasonal framework, spanning 52 or 53 weeks, beginning in week 36 of the current year and ending in week 35 of the subsequent year.”

Your interpretation of Figure 1 is:

“This cyclical pattern was disrupted during the COVID- 19 pandemic (2020-2022), with influenza activity experiencing an unprecedented decline. Influenza cases were markedly suppressed during the pandemic years, coinciding with the implementation of widespread nonpharmaceutical interventions (NPIs) such as mask wearing, physical distancing, and international travel restrictions. It is worth noting that there was a decrease in testing for Influenza and other viruses, like dengue [13], during the COVID-19 pandemic, which might have led to underreporting of cases [11]. The containment and health index, plotted alongside influenza cases, reflects the intensity of these measures, which peaked in early 2020. In the post-pandemic period (2023-2024), influenza activity exhibited a robust rebound, returning to levels consistent with the pre-pandemic era. This resurgence underscores the waning impact of NPIs as public health measures were relaxed and global travel resumed. The results highlight how behavioral and policy driven interventions, though effective in the short term, are not sufficient to sustain longterm suppression of respiratory pathogens like influenza. The geographic distribution of influenza cases across U.S. states over six influenza seasons illustrates the differential impact of the pandemic and subsequent recovery periods across regions (Figure 2). Additionally, a correlation analysis indicated that states in the U.S. with higher flu incidence experienced more COVID-19 cases during and after the pandemic period (Supplementary Figure S1). This indicates that the common risk factors, including population density, healthcare accessibility, and behavioral determinants, may affect the transmission dynamics of these respiratory viruses [14] “

[It is easy to see the missing peak in the 2020-2021 season but then subsequent peaks are higher]

[Could you please statistically analyses the data in Figure 1. Seasonal influenza and COVID-19 dynamics, Figure 2. Geographic trends, Figure 3. Mortality and Figure 4. Vaccination coverage to provide detailed statistical support for your analyses. If not included in your team you may need to enlist a statistician experienced in trend analyses].

Author Response

This is an important article. However, it lacks appropriate statistical analyses.

Reply: We thank the reviewer for the acknowledgment and constructive feedback. We added appropriate statistical analysis to support our results.

You wrote: “The influenza surveillance period follows a seasonal framework, spanning 52 or 53 weeks, beginning in week 36 of the current year and ending in week 35 of the subsequent year.”

Your interpretation of Figure 1 is:

“This cyclical pattern was disrupted during the COVID- 19 pandemic (2020-2022), with influenza activity experiencing an unprecedented decline. Influenza cases were markedly suppressed during the pandemic years, coinciding with the implementation of widespread nonpharmaceutical interventions (NPIs) such as mask wearing, physical distancing, and international travel restrictions. It is worth noting that there was a decrease in testing for Influenza and other viruses, like dengue [13], during the COVID-19 pandemic, which might have led to underreporting of cases [11]. The containment and health index, plotted alongside influenza cases, reflects the intensity of these measures, which peaked in early 2020. In the post-pandemic period (2023-2024), influenza activity exhibited a robust rebound, returning to levels consistent with the pre-pandemic era. This resurgence underscores the waning impact of NPIs as public health measures were relaxed and global travel resumed. The results highlight how behavioral and policy driven interventions, though effective in the short term, are not sufficient to sustain longterm suppression of respiratory pathogens like influenza. The geographic distribution of influenza cases across U.S. states over six influenza seasons illustrates the differential impact of the pandemic and subsequent recovery periods across regions (Figure 2). Additionally, a correlation analysis indicated that states in the U.S. with higher flu incidence experienced more COVID-19 cases during and after the pandemic period (Supplementary Figure S1). This indicates that the common risk factors, including population density, healthcare accessibility, and behavioral determinants, may affect the transmission dynamics of these respiratory viruses [14] “

Reply: We thank the reviewer for nicely summarizing our study findings.

[It is easy to see the missing peak in the 2020-2021 season, but then subsequent peaks are higher]

Reply: Thank you for pointing out the discrepancy. We modified the results about the higher picks of flu cases in the post-pandemic period.

[Could you please statistically analyses the data in Figure 1. Seasonal influenza and COVID-19 dynamics, Figure 2. Geographic trends, Figure 3. Mortality and Figure 4. Vaccination coverage to provide detailed statistical support for your analyses. If not included in your team you may need to enlist a statistician experienced in trend analyses].

Reply: Thank you for your valuable feedback. We have conducted statistical analysis to support our findings. We performed an Interrupted Time Series (ITS) analysis to assess changes in weekly influenza case trends across the pre-pandemic, pandemic, and post-pandemic periods. Additionally, we applied the Wilcoxon signed-rank test to evaluate changes in influenza cases and vaccination coverage between flu seasons across U.S. states. We have incorporated these analyses into the Methods and Results sections with supplementary figures and revised the manuscript accordingly.

Reviewer 2 Report

Comments and Suggestions for Authors

This manuscript provided interesting epidemic data about influenza infected case in before, during and past of COVID-19 pandemic and wished to conclude the interaction of these two respiratory pathogens. Such a analysis could be atractive for readers and provide useful data for the epidemic analysis on respiratory infectious diseases. It could be considered for its publication after its improved. Suggestions for its modificatio as below:

1: The aim of this study was to find the relationship of influenza and SARS-CoV-2 as their pandemics observed in commen space and time, therefore, it should focus on the some critical factors, as like geographic factor that was involved in manuscript but no conclusion, public measures that was involved also without further analysis.

2: The figure 1 of manuscript was interesting data, however, it could be better if it could present complete data of COVID-19 in post pandemic and showed comparison of patients peak of influenza and COVID-19, because there were no control of  public measures in post pandemic.

3: As all of us know, it could not be easy to get enough data about these two diseases, and, to investigate all possible factors related to the interaction of these two pathogens could be difficult. In this case, that the manuscript could be modified into a brief report might be a choose.

Author Response

This manuscript provided interesting epidemic data about influenza infected case in before, during and past of COVID-19 pandemic and wished to conclude the interaction of these two respiratory pathogens. Such an analysis could be attractive for readers and provide useful data for the epidemic analysis on respiratory infectious diseases. It could be considered for its publication after its improved. Suggestions for its modification as below:

Comment 1: The aim of this study was to find the relationship of influenza and SARS-CoV-2 as their pandemics observed in commen space and time, therefore, it should focus on the some critical factors, as like geographic factor that was involved in manuscript but no conclusion, public measures that was involved also without further analysis.

Reply:  We appreciate the reviewer’s insights and acknowledge the complexity of addressing all critical factors influencing the interaction between influenza and SARS-CoV-2. Our primary objective was to illustrate the shifts in influenza circulation within the broader context of the COVID-19 pandemic, particularly in relation to containment measures and public health interventions. To enhance clarity, we have refined Figure 1 to clearly delineate pre-pandemic, pandemic, and post-pandemic phases, allowing for a more structured interpretation of influenza trends. The figure highlights a marked suppression of influenza cases during the peak of COVID-19 containment efforts, followed by a resurgence as restrictions eased. This pattern underscores the influence of nonpharmaceutical interventions, altered virus circulation dynamics, and vaccination efforts. We have conducted statistical analysis to support our findings. In response to this feedback, we have further elaborated on these findings in the Results and Conclusion sections to better contextualize the observed epidemiological patterns.

Comment 2: The figure 1 of manuscript was interesting data, however, it could be better if it could present complete data of COVID-19 in post pandemic and showed comparison of patients peak of influenza and COVID-19, because there were no control of public measures in post pandemic.

Reply: We thank the reviewer for the suggestion. We have now redrafted the figure for better clarity. However, the data from OWID (https://ourworldindata.org/covid-cases) for the US are available from January 9, 2020, to May 21, 2023. After this date, there are no reported COVID-19 cases (https://ourworldindata.org/covid-jhu-who).

Comment 3: As all of us know, it could not be easy to get enough data about these two diseases, and, to investigate all possible factors related to the interaction of these two pathogens could be difficult. In this case, that the manuscript could be modified into a brief report might be a choose.

Reply: We appreciate the reviewer's suggestion and acknowledge the challenges in obtaining comprehensive data and investigating all possible interacting factors. However, as this manuscript was invited as an article by the journal, we cannot modify its format. To strengthen our findings, we have conducted additional statistical analyses and further refined our discussion.

Reviewer 3 Report

Comments and Suggestions for Authors

The interplay between the non-pharmaceutical interventions (NPI) mandated and practiced during the SARS-CoV-2 pandemic and the incidence (case ascertainment, hospital admissions, mortality) of other respiratory and enteric infections has been described previously. In that sense, this manuscript does not provide novel information. On the other hand what this manuscript provides is additional data for the more recent infection seasons (especially 2022-2023 and 2023-2024).

I have some comments for the authors to consider and some suggestions for improvement of the manuscript.

1. Line 107. Influenza cases rebounded in the 2022-2023 seasons as well as in the 2023-2024 seasons. I think you will find that the initial peak coincided approximately with the relaxation of NPI practiced during the SARS-CoV-2 pandemic. This should be mentioned at this point and see also next comment.

2. Figure 1. I think a more telling and informative figure would include not the WHO declarations of pandemic start and end but rather times of the implementation of NPI and the relaxation of NPI. See Sullivan et al. Euro Surveil. 2020;25(47):2001847, Mattana et al.  J Global Antimicrob Res. 2022:S2213-7165, Ijaz et al. Microbiol Open Access 2022, 8:229. Any reason why you have chosen not to reference these papers?

3. Figure 3. This figure is made unnecessarily confusing by the use of different density keys for the different seasons. I recommend that the same key be applied to each season, then comparisons will be much easier for your readers.

4. Line 138. Florida displayed significant activity during the 2020-2021 season (similar in fact to 2019-2020, per the figure)

5. Line 142. The shift was particularly to California. Any speculation as to why?

6. Figure 3. Please consider keeping the density key the same for each season. 

7. Lines 157 to 170. Discussion of deaths in the 2020-2021 season mirroring case counts (line 157-158 is hindered by the fact that no death data are shown for the 2020-2021 season. This comment applies also to line 161. Line 164. The plots do not show much variability in mortality to me, but again the use of different density keys may make this difficult for the reader to see.

8. Line 168-170. I think this is a reach. The findings show that influenza mortality decreases along with case counts during the pandemic. Not sure how the findings relate to effective surveillance, timely intervention, and vaccination programs.

9. Figure 4. Please keep the density key consistent for the different seasons.

10. Lines 200-204. Upon what sources do you make the statement regarding epidemiology of other pathogens (i.e., other than SARS-CoV-2 and influenza) during the pandemic?

11. Lines 209-213. Again, Figure 3 does not display mortality data for the 2020-2021 season, so on what basis do you make the statement on relationship between NPI and mortality? This comment applies also to lines 238-239.

12. Line 229-230. The resurgence in influenza cases is not really a limitation of the behavioral modifications. Rather it is a limitation in maintaining the NPI following the pandemic.

Author Response

The interplay between the non-pharmaceutical interventions (NPI) mandated and practiced during the SARS-CoV-2 pandemic and the incidence (case ascertainment, hospital admissions, mortality) of other respiratory and enteric infections has been described previously. In that sense, this manuscript does not provide novel information. On the other hand what this manuscript provides is additional data for the more recent infection seasons (especially 2022-2023 and 2023-2024).

I have some comments for the authors to consider and some suggestions for improvement of the manuscript.

  1. Line 107. Influenza cases rebounded in the 2022-2023 seasons as well as in the 2023-2024 seasons. I think you will find that the initial peak coincided approximately with the relaxation of NPI practiced during the SARS-CoV-2 pandemic. This should be mentioned at this point and see also next comment.

Reply: Thank you for your observation. We have noted that the initial peak in influenza cases during the 2022-2023 and 2023-2024 seasons coincided with the relaxation of non-pharmaceutical interventions (NPI) implemented during the SARS-CoV-2 pandemic. This has been mentioned in the revised manuscript for clarity.

  1. Figure 1. I think a more telling and informative figure would include not the WHO declarations of pandemic start and end but rather times of the implementation of NPI and the relaxation of NPI. See Sullivan et al. Euro Surveil. 2020;25(47):2001847, Mattana et al.  J Global Antimicrob Res. 2022:S2213-7165, Ijaz et al. Microbiol Open Access 2022, 8:229. Any reason why you have chosen not to reference these papers?

Reply: Thank you for your insightful suggestion. We have added federal initiatives as state-specific data would be too complex to represent in this graph. Additionally, we have included the relevant references you suggested in the manuscript

  1. Figure 3. This figure is made unnecessarily confusing by the use of different density keys for the different seasons. I recommend that the same key be applied to each season, then comparisons will be much easier for your readers.

Reply: We thank the reviewer for raising this point. We have now standardized the legend across all seasons in Figure 3 to make comparisons easier for our readers.

  1. Line 138. Florida displayed significant activity during the 2020-2021 season (similar in fact to 2019-2020, per the figure)

Reply: We have standardized the legend across all seasons in Figure 2 to facilitate easier comparisons for our readers. We apologize for any confusion this may have caused.

  1. Line 142. The shift was particularly to California. Any speculation as to why?

Reply: California experienced higher influenza transmission in the 2022-2023 season due to the relaxation of COVID-19 measures, concurrent circulation of multiple viruses, and high population density.

https://www.cdph.ca.gov/Programs/CID/DCDC/CDPH%20Document%20Library/Immunization/Week2022-2348_FINALReport.pdf; https://www.cdph.ca.gov/Programs/CID/DCDC/CDPH%20Document%20Library/Immunization/Week2022-2344_FINALReport.pdf.

  1. Figure 3. Please consider keeping the density key the same for each season. 

Reply: We thank the reviewer for raising this point. We have now standardized the legend across all seasons in Figure 3 to make comparisons easier for our readers.

  1. Lines 157 to 170. Discussion of deaths in the 2020-2021 season mirroring case counts (line 157-158 is hindered by the fact that no death data are shown for the 2020-2021 season. This comment applies also to line 161. Line 164. The plots do not show much variability in mortality to me, but again the use of different density keys may make this difficult for the reader to see.

Reply: We have now standardized the legend across all seasons in Figure 3 to make comparisons easier for our readers. We have revised the text to eliminate any ambiguity and ensure clarity.

  1. Line 168-170. I think this is a reach. The findings show that influenza mortality decreases along with case counts during the pandemic. Not sure how the findings relate to effective surveillance, timely intervention, and vaccination programs.

Reply: Thank you for your feedback. We understand your concern. We have revised the text to better explain how the decrease in influenza mortality and case counts during the pandemic is linked to effective surveillance, timely intervention, and vaccination programs. These measures played a crucial role in monitoring and controlling the spread of influenza, thereby reducing both cases and mortality rates.

  1. Figure 4. Please keep the density key consistent for the different seasons.

Reply: We thank the reviewer for raising this point. We have now standardized the legend across all seasons in Figure 4 to make comparisons easier for our readers.

  1. Lines 200-204. Upon what sources do you make the statement regarding epidemiology of other pathogens (i.e., other than SARS-CoV-2 and influenza) during the pandemic?

Reply: We thank the reviewer for highlighting this point. We have now included the appropriate references showing the similar pattern of influenza during the SARS CoV-2 pandemic.

  1. Lines 209-213. Again, Figure 3 does not display mortality data for the 2020-2021 season, so on what basis do you make the statement on relationship between NPI and mortality? This comment applies also to lines 238-239.

Reply: We thank the reviewer for raising this point. We have made the necessary edits. The previous mention of influenza death data was incorrect and should have referred to COVID-19 mortality. We have corrected this within the text.

  1. Line 229-230. The resurgence in influenza cases is not really a limitation of the behavioral modifications. Rather it is a limitation in maintaining the NPI following the pandemic.

Reply: Thank you for pointing this out. We have rephrased the sentence to clarify that the resurgence in influenza cases is due to the challenge of maintaining non-pharmaceutical interventions (NPI) following the pandemic, rather than a limitation of the behavioral modifications themselves.

Round 2

Reviewer 1 Report

Comments and Suggestions for Authors

Unfortunately the article was presented with the changes made in blue and figures in the margin and was difficult to read. Was this done because the authors did not want to provide a narrative account and explanation of the changes they made?

  1. “It is worth noting that there was a decrease in testing for Influenza and other viruses, like dengue [13], during the COVID-19 pandemic, which might have led to underreporting of cases [11].”

[General statements like this have been widely made in the literature. Please provide numerical data to support your observations. This will be very helpful to readers].

55 “The pandemic disrupted the global circulation of seasonal influenza, partly due to changes in air travel connectivity between world regions.”

[General statements like this have been widely made in the literature. Please provide numerical data on air travel especially between crucial regions to support your observations.]

“Moreover, the initial decline in flu activity was strongly influenced by the global adoption of stringent public health measures including mask mandates and physical distancing [9–11], which effectively curtailed person-to-person transmission of respiratory viruses. Since the emergence of SARS-CoV-2 in December 2019 it has infected over 700 million individuals globally and caused more than seven million deaths.”

[please provide numerical data to support these statements]

“The United States alone, was one of the most affected countries, with CDC estimates bordering 40 million COVID-19 infections occurred up to 2024 [12].”

[Please provide numerical data why the US was so badly affected].

[Please provide an explanation for the correlations in Supplementary Figure S2]

Two figures appear to be labelled Supplemental Fig S3

Supplementary Figure S3.

[Please provide explanation of results for this box plot]

Supplemental Figure S3

[please explain why no data are available for two of the periods of interest]

Typos

120 picks compared (change to peaks)

Author Response

Comment 1: Unfortunately the article was presented with the changes made in blue and figures in the margin and was difficult to read. Was this done because the authors did not want to provide a narrative account and explanation of the changes they made?

Reply: We acknowledge the reviewer’s concern regarding the readability of the tracked changes. The blue color was automatically assigned by Word, and the standard review practice includes marking text and figure modifications to transparently reflect the revisions made in response to comments. Additionally, we provided a detailed narrative of the changes implemented. We regret any inconvenience caused and trust that our responses adequately clarify the modifications. Should further clarification be required, we are happy to provide additional details.

Comment 2: 113.“It is worth noting that there was a decrease in testing for Influenza and other viruses, like dengue [13], during the COVID-19 pandemic, which might have led to underreporting of cases [11].”

[General statements like this have been widely made in the literature. Please provide numerical data to support your observations. This will be very helpful to readers].

Reply: We acknowledge the reviewer’s request for numerical data to support this statement. Unfortunately, we do not have specific quantitative data on the reduction in testing during the COVID-19 pandemic. However, this observation is well-documented in the literature, as cited in now references [15,16,17]. If needed, we can refine the statement to clarify that this is based on prior studies rather than primary data.

Comment 3: 55 “The pandemic disrupted the global circulation of seasonal influenza, partly due to changes in air travel connectivity between world regions.”

[General statements like this have been widely made in the literature. Please provide numerical data on air travel especially between crucial regions to support your observations.]

Reply: During the pandemic, a worldwide air travel ban was implemented, significantly disrupting global mobility, including the circulation of seasonal influenza. This is a well-documented in several previous publications. Here, we have included relevant citations to support this statement. We feel that this substantiates our statements regarding the decrease in mobility.

Comment 4: “Moreover, the initial decline in flu activity was strongly influenced by the global adoption of stringent public health measures including mask mandates and physical distancing [9–11], which effectively curtailed person-to-person transmission of respiratory viruses. Since the emergence of SARS-CoV-2 in December 2019 it has infected over 700 million individuals globally and caused more than seven million deaths.”

[please provide numerical data to support these statements]

Reply: The impact of public health measures on flu activity is well-documented, and we provide several citations to support this statement. As highlighted in the introduction, the 2020–2021 flu season in the US was the lowest on record since the CDC began its current surveillance system in 1997, despite extensive testing for upper respiratory viruses. This substantial decline aligns with the widespread adoption of non-pharmaceutical interventions including mask wearing, decreased mobility, and social distancing.

Comment 5: “The United States alone, was one of the most affected countries, with CDC estimates bordering 40 million COVID-19 infections occurred up to 2024 [12].”

[Please provide numerical data why the US was so badly affected].

Reply: We have revised the text to clarify the points raised.

Comment 6: [Please provide an explanation for the correlations in Supplementary Figure S2]

Reply: We have improved the results of Supplementary Figure S2.

Comment 7: Two figures appear to be labelled Supplemental Fig S3

Reply: We cited Supplemental Figure 3 twice in the text; however, there is only one Supplemental Figure 3. To improve clarity, we have labeled the two panels as “a” and “b”.

Comment 8: Supplementary Figure S3.

[Please provide explanation of results for this box plot]

Reply: We have provided additional details on these results.

Comment 9: Supplemental Figure S3

[please explain why no data are available for two of the periods of interest]

Reply: In Supplemental Figure S3, there are no missing periods of interest; the reviewer likely referred to Figure 3. Unfortunately, the CDC data website does not provide influenza-related mortality data for these two influenza seasons. They do not provide clarification.

Comment 10: Typos

120 picks compared (change to peaks)

Reply: Corrected.

Reviewer 3 Report

Comments and Suggestions for Authors

thank you for the replies to my comments and for making the suggested changes.

I will now recommend acceptance, however, please check the legend for Supplementary Figure S1 as it looks like some of the symbols/colors mentioned in the legend do not align with the key in the figure itself

Author Response

thank you for the replies to my comments and for making the suggested changes.

I will now recommend acceptance, however, please check the legend for Supplementary Figure S1 as it looks like some of the symbols/colors mentioned in the legend do not align with the key in the figure itself

Reply:  Thank you for your insightful comment. We have revised the figure legend to ensure it aligns cohesively with the figure
